# A Mouse Model to Assess STAT3 and STAT5A/B Combined Inhibition in Health and Disease Conditions

**DOI:** 10.3390/cancers11091226

**Published:** 2019-08-22

**Authors:** Herwig P. Moll, Julian Mohrherr, Leander Blaas, Monica Musteanu, Patricia Stiedl, Beatrice Grabner, Katalin Zboray, Margit König, Dagmar Stoiber, Thomas Rülicke, Sabine Strehl, Robert Eferl, Emilio Casanova

**Affiliations:** 1Department of Physiology, Center of Physiology and Pharmacology, Comprehensive Cancer Center (CCC), Medical University of Vienna, 1090 Vienna, Austria; 2Ludwig Boltzmann Institute for Cancer Research (LBI-CR), 1090 Vienna, Austria; 3Department of Biosciences and Nutrition, Center for Innovative Medicine, Karolinska Institutet, Novum, 14183 Huddinge, Sweden; 4CNIO (Spanish National Cancer Research Centre), E-28029 Madrid, Spain; 5Plant Protection Institute, Centre for Agricultural Research, Hungarian Academy of Sciences, 2462 Martonvásár, Hungary; 6Children’s Cancer Research Institute, St. Anna Kinderkrebsforschung, 1090 Vienna, Austria; 7Department of Pharmacology, Center of Physiology and Pharmacology, Comprehensive Cancer Center (CCC), Medical University of Vienna, 1090 Vienna, Austria; 8Institute of Laboratory Animal, Science, University of Veterinary Medicine Vienna, 1210 Vienna, Austria; 9Institute of Cancer Research, Medical University Vienna & Comprehensive Cancer Center (CCC), 1090 Vienna, Austria

**Keywords:** bacterial artificial chromosome, recombineering, Cre/loxP, gene targeting, embryonic stem cells, liver steatosis

## Abstract

Genetically-engineered mouse models (GEMMs) lacking diseased-associated gene(s) globally or in a tissue-specific manner represent an attractive tool with which to assess the efficacy and toxicity of targeted pharmacological inhibitors. *Stat3* and *Stat5a/b* transcription factors have been implicated in several pathophysiological conditions, and pharmacological inhibition of both transcription factors has been proposed to treat certain diseases, such as malignancies. To model combined inhibition of *Stat3* and *Stat5a/b* we have developed a GEMM harboring a flox *Stat3*-*Stat5a/b* allele (*Stat5/3^loxP/loxP^* mice) and generated mice lacking *Stat3* and *Stat5a/b* in hepatocytes (*Stat5/3^Δhep^*^/*Δhep*^). *Stat5/3^Δhep/Δhep^* mice exhibited a marked reduction of STAT3, STAT5A and STAT5B proteins in the liver and developed steatosis, a phenotype that resembles mice lacking *Stat5a/b* in hepatocytes. In addition, embryonic deletion of *Stat3* and *Stat5a/b* (*Stat5/3^Δ^*^/*Δ*^ mice) resulted in lethality, similar to *Stat3^Δ^*^/*Δ*^ mice. This data illustrates that *Stat5/3^loxP/loxP^* mice are functional and can be used as a valuable tool to model the combined inhibition of *Stat3* and *Stat5a/b* in tumorigenesis and other diseases.

## 1. Introduction

Genetically-engineered mouse models (GEMMs) lacking gene(s) of interest in the whole animal or in a tissue-specific manner offer a unique opportunity to model the effects of pharmacological inhibitors, as well as to assess toxicity in models of health and disease [1]. Upon identification of a cancer target gene(s), the toxicity and efficacy of future inhibitors can be evaluated by crossing mice harboring alleles of interest flanked by loxP sites (flox mice) with tissue-specific and inducible Cre recombinase expressing mouse strains and deleting the gene(s) in appropriate cancer-mouse models [2]. This strategy offers a straight forward and rigorous method for target validation before undertaking costly and time-consuming development of antitumorigenic targeted inhibitors.

Signal transducer and activator of transcription family members (STAT1, 2, 3, 4, 5A, 5B and 6) are transcription factors which are activated in response to multiple cytokines and growth factors. The binding of ligands to their cognate receptors results in the activation of Janus kinase (JAK) family members (JAK1, 2, 3 and Tyrosine kinase 2, TYK2). Tyrosine phosphorylation of STATs by JAK family members induces homo- and hetero- dimerization, as well as subsequent translocation into the nucleus where STAT proteins bind to their specific DNA sequences and control the transcription of multiple genes [3,4]. STAT3, STAT5A and STAT5B have been implicated in several physiological and disease processes such as cell division, migration, survival, cellular responses to pathogens, breast development, inflammation and cancer [5]. Indeed, the over-activation of STAT3 and STAT5A/B has been reported in human hematopoietic and solid malignancies, suggesting that these transcription factors behave as oncogenic drivers in those tumors [6,7]. These observations have been complemented with functional studies using GEMMs harboring flox alleles for *Stat3* [8,9,10,11,12,13] and *Stat5a/b* [14] genes. Consequently, several research groups have developed inhibitors targeting STAT3 or STAT5A/B to treat malignancies [6]. Unfortunately, monotherapy treatment of tumors almost invariably results in tumor resistance. Tumor cells acquire resistance to drugs by (re)activating alternative or redundant signaling pathways [15]. STAT3 and STAT5A/B have overlapping functions in several (patho)physiological contexts [16,17,18,19]; thus it is expected that tumor cells will acquire resistance to single inhibitors by compensatory mechanisms relaying in the activation of the non-inhibited STAT family member. Therefore, dual inhibitors targeting both STAT3 and STAT5A/B or a combination of single STAT3 and STAT5A/B inhibitors may be more effective [20], albeit perhaps also resulting in increased toxicity. *Stat3*, *Stat5a* and *Stat5b* are genetically-linked genes, clustering within a 160 kb genomic region of mouse chromosome 11. This should preclude modeling simultaneous inhibition of STAT3 and STAT5A/B simply by intercrossing *Stat3* and *Stat5a/b* single flox mice. To overcome this issue, we have generated a new GEMM which contains an allele harboring the *Stat3*, *Stat5a* and *Stat5b* genes flanked by loxP sites (*Stat5/3^loxP/loxP^* mice). *Stat5/3^loxP/loxP^* mice crossed to a transgenic line expressing the Cre recombinase in hepatocytes showed deletion of STAT3 and STAT5A/B in the liver and developed steatosis at the age of eight weeks. Furthermore, embryonic deletion of STAT3 and STAT5A/B resulted in a lethal phenotype. This demonstrates that the *Stat5/3^loxP/loxP^* mouse is functional and can be readily used to model STAT3 and STAT5A/B double inhibition in cancer and other disease-experimental mouse models.

## 2. Results and Discussion

### 2.1. Generation of Stat3 and Stat5a/b Flox (Stat5/3^loxP/loxP^) Mice

A Bacterial Artificial Chromosome (BAC)-based targeting construct [21,22] spanning the *Stat3*, *Stat5a* and *Stat5b* locus was generated as follows: a 200 kb BAC (BAC, RPCI-23-362J7) (Figure 1A and Appendix A) was modified by introducing a loxP-FRT3-Neomycin-FRT3 cassette using BAC homologous recombination [23] upstream of the *Stat5b* gene as previously described [14]. In a second recombination step, a FRT-Hygromycin-FRT-loxP cassette was introduced within the first intron of the *Stat3* gene between exons 1 and 2. *Stat3* exon 2 contains the ATG translation initiation codon; thus, the flox strategy predicts the absence of a STAT3 truncated protein (Appendix A). The BAC-based targeting construct was linearized using the restriction enzyme *Not I* and electroporated into HM-1 mouse embryonic stem cells (ES, 129/ola origin [24]). Out of around thirty ES cell clones which survived the neomycin/hygromycin double selection, eighteen appeared to have an undifferentiated morphology and were chosen for further analysis. We performed Southern blot analysis of genomic DNA derived from the selected clones using a DNA probe upstream of the hygromycin cassette to visualize the endogenous (wt) and the transgenic alleles (Figure 1B). We presumed that equal intensities of wt and transgene bands either indicated correctly targeted ES cells which had incorporated the transgene in the endogenous *Stat5/3* locus or random integration of two copies of the transgene. Unequal intensities between the wt and transgene bands would suggest random integration of the transgene into the genome. Clones 9, 10, 11, 13 and 15 (Figure 1B) showed wt and transgene bands of approximately equal intensities and were further evaluated. 

We performed Fluorescence In Situ Hybridization (FISH) using the RPCI-23-362J7 BAC as probe for the selected clones (Figure 2A and Table 1). wt ES cells showed mainly two signals, but we also observed nuclei containing three signals, suggesting chromosomal abnormalities. Notably, mouse ES cells tend to harbor a trisomy of chromosome 11 [25], which harbors the *Stat5/3* locus. Clone 8 (which was included in the analysis as a control because the transgene band showed a higher intensity than the wt in the Southern blot, indicating a random integration event) displayed three signals. Clones 10, 11, 13 and 15 also showed mainly three or four signals (Table 1), suggesting that the transgene had been randomly incorporated into the genome of these clones. In contrast, clone 9 exhibited two signals, indicating correct recombination of the transgene into the endogenous locus. Furthermore, PCR analysis of genomic DNA showed that the 5′ and 3′ ends of the BAC construct were not present in clone 9, demonstrating that they were lost during homologous recombination (Figure 2B). Additionally, genomic qPCR using three pairs of primers flanking either the neomycin cassette (P3 and P4), a central region in the *Stat5a* locus (P5 and P6) or the hygromycin cassette (P7 and P8) confirmed that clone 9 contained one copy of the wt endogenous *Stat5/3* locus and one copy of the transgene (Figure 2C). Of note, primers P3/P4 and P7/8 do not amplify the transgenic construct. Finally, the functionality of the loxP sites was assessed by transfecting a plasmid expressing the Cre recombinase [26] into clone 9. Indeed, deletion of the *Stat5/3* locus was achieved as demonstrated by PCR using genomic DNA and primers P9 and P10 (Figure 2D). Therefore, we used ES cell clone 9 for the generation of chimeric animals via injection of the ES cell clone into blastocysts. Subsequently, these mice were mated to C57BL/6N mice to establish a mouse line harboring the *Stat5/3^loxp^* allele.

### 2.2. Functional Validation of Stat5/3^loxP/loxP^ Mice

*Stat5/3^loxP/loxP^* mice are viable; genotyping shows that they are born at Mendelian ratio (Figure 3A and Table 2) and do not exhibit any obvious phenotype. *Stat5/3^loxP/loxP^* mice (backcrossed five generations into C57BL/6N) were crossed to a transgenic line expressing the Cre recombinase under control of the *Actb* promoter [27] to obtain a null allele (*Stat5/3^Δ/+^* mice). Intercrossing of *Stat5/3^Δ/+^* mice did result in viable *Stat5/3^Δ^*^/*Δ*^ blastocysts, but not in living offspring (Table 2), suggesting that *Stat5/3^Δ^*^/*Δ*^ mice were embryonic lethal, as previously described for *Stat3^Δ^*^/*Δ*^ mice [28].

Previous reports have shown that the deletion of *Stat5a/b* in hepatocytes results in liver steatosis [29]. Therefore, we investigated the effect of concomitant *Stat5a/b* and *Stat3* deletion in the liver. *Stat5/3^loxP/loxP^* mice were intercrossed with transgenic mice expressing the Cre recombinase under the control of the tetO promoter (tetO-Cre mice [30]) and with mice expressing the tetracycline-controlled transactivator (tTA) in hepatocytes from the liver-enriched activator protein promoter (LAP-tTA mice [31]) to obtain triple transgenic animals: *Stat5/3^loxP/loxP^*; LAP-tTA; tetO-Cre (hereby *Stat5/3^Δhep/Δhep^* mice, Δ*hep*: deleted in hepatocytes). Liver-specific deletion of the *Stat5/3* locus was confirmed by PCR using genomic DNA from different organs of *Stat5/3^Δhep/Δhep^* mice (Figure 3B). Furthermore, Western blot analysis revealed that STAT5A/B and STAT3 proteins were absent in livers of *Stat5/3^Δhep/Δhep^* mice (Figure 3C). At the macroscopic level, *Stat5/3^Δhep/Δhep^* mouse livers were enlarged and pale, suggesting lipid accumulation (Figure 3D). HE-stained sections from livers of *Stat5/3^Δhep/Δhep^* mice confirmed a steatotic phenotype. Taken together, these results prove that *Stat5/3^loxP/loxP^* mice are functional and recapitulate the liver steatotic phenotype found in mice lacking *Stat5a/b* in hepatocytes [29].

Modeling STAT3 and STAT5A/B double inhibition using GEMM in cancer and other diseases is an attractive strategy to assess inhibitor efficacy and side effects. *Stat5a/b* and *Stat3* are genetically linked and located within an approximately 160 kb region of the mouse genome. Consequently, combined deletion of *Stat5a/b* and *Stat3* by breeding currently-available mice harboring conditional alleles for *Stat5a/b* and *Stat3* would be a challenging task. On the other hand, Singireddy and colleagues have reported the generation of double *Stat5a/b* and *Stat3* flox mice by breeding individual flox mice; they found that double *Stat5a/b* and *Stat3* flox mice were born at expected Mendelian ratios [32]. This may indicate that the *Stat5a/b* and *Stat3* genomic region harbors a meiotic recombination hotspot whose efficiency could be genetic background-dependent [33]. We have used a different approach and generated a new transgenic mouse line carrying the *Stat5a*, *Stat5b* and *Stat3* genes flanked by loxP sites. We have used a BAC-based targeting construct to modify ES cells [21,22]. This strategy has the advantage to flox relatively large genomic regions with one single targeting event in ES cells. Consequently, in contrast to the Southern blot strategy used for the detection of conventional targeting constructs [34], correctly targeted ES cells need to be identified by FISH and/or quantification of the endogenous and the transgenic alleles by qPCR [22]. Since *Stat5/3^loxP/loxP^* mice were generated using ES derived from the 129/ola strain, we backcrossed these mice for five generations into C57BL/6N, a widely-used experimental mouse genetic background. Nevertheless, it is expected that the genomic region around the *Stat5a/b* and *Stat3* genes is flanked by 129/ola ES cell derived DNA. This 129/ola genomic region may contain mutations that confound the interpretation of the phenotypes found upon the loss of *Stat5a/b* and *Stat3* genes [35]. This issue could be overcome by generating the mice directly in C57BL/6N-derived ES cells [36]. It is known that increased distances between the loxP sites have been correlated with lower recombination efficiency [37]. Therefore, we aimed to demonstrate the functionality of *Stat5/3^loxP/loxP^* mice by deleting the flox region in hepatocytes. Indeed, despite the relatively large genomic region (135 kb) flanked by loxP sites, we were able to achieve an efficient deletion of the *Stat5a/b* and *Stat3* genes using a mouse line expressing the Cre recombinase in the liver under the control of the tetracycline-transactivator [30,31]. However, other Cre transgenic lines may not be that efficient. In this sense, crossing *Stat5/3^loxP/loxP^* mice with Alfp Cre mice (a transgenic line which expresses the Cre recombinase in hepatocytes and cholangiocytes starting at embryonic day 9.5) [38] resulted in incomplete recombination.

Although loss-of-function approaches using knockout mice may serve to validate targets on oncology, it has also some disadvantages. Due to the redundancy in biology, upon gene deletion, it is not usual that other proteins with similar functions become activated, resulting in compensatory mechanisms which may confound the interpretation phenotype [39]. In this sense, it has been reported that deletion of *Stat5a/b* (but not *Stat3*) in hepatocytes results in liver steatosis [29], possibly due to an unphysiological activation of STAT3 as cellular response to the lack of STAT5/B. Interestingly, in our model, the deletion of *Stat5a/b* and *Stat3* in hepatocytes results in liver steatosis. This finding not only validates the functionality of our mouse model, but also suggests that the steatotic phenotype observed upon deletion of *Stat5a/b* in hepatocytes is not caused by aberrant-activation/regulation of *Stat3*.

## 3. Material and Methods

### 3.1. BAC Recombineering

*E. coli* DH10B bacteria harboring the RPCI-23-362J7 BAC (purchased from CHORI) were electroporated with the pSC101-BAD-abgA plasmid [40] and grown at 30 °C until the OD reached 0.2. L-Arabinose was then added to the culture (0.2% final concentration) and cells were shifted to 37 °C until the OD reached 0.5. Cells were harvested and electrocompetent cells were prepared. A neomycin/kanamycin selection cassette flanked by FRT3 sites and harboring a 5′ loxP site was recombined in the BAC upstream of *Stat5b* by electroporating a DpnI-digested PCR product amplified from a PGK/Tn5-neomycin cassette [41] using the following primers: *CCAGAGAACCTGTGGGCAATGGAATGGGCAGAAGCCTCAACCTACACTGC***ATAACTTCGTATAATGTATGCTATACGAAGTTAT**GAAGTTCCTATACTATTTGAAGAATAGGAACTTCGTGGAGTCGAGGAATTCTACC and *TGAGAGGAAAGCATGAAAGGGTTGGAGCCAGGGCATTATGTGTGAGGCAG*GAAGTTCCTATTCTTCAAATAGTATAGGAACTTCCAAAAACCAACACACAGATCATG (Italics: homology region to the BAC, bold: loxP site, underlined: FRT3 site). Putative recombinant clones were selected using kanamycin (25 μg/mL). Correctly recombined BACs were validated by PCR and direct BAC sequencing. In a second round of recombineering, a hygromycin selection cassette flanked by FRT sites harboring a 3′ loxP site was recombined into the BAC containing the neomycin/kanamycin resistance cassette by electroporating a Dpn I-digested PCR product amplified from a pUBC/EM7-hygromicin plasmid [42] using the following primers: *GTCTGTAACTCAAGAGGCTTGTTTCAGTGTTGATGTTTGGTGTATTTCGC*GAAGTTCCTATTCTCTAGAAAGTATAGGAACTTCGGCCTCCGCGCCGGGTTTTGGC and *GTTCTTGATGGTCCAGTTAATTGGAAAAAGGCACAAGCTGGTCTCTGGTT***ATAACTTCGTATAGCATACATTATACGAAGTTAT**GAAGTTCCTATACTTTCTAGAGAATAGGAACTTCCCTGCAACATGAATATTAGAA (Italics: homology region to the BAC, bold: loxP site, underlined: FRT site). Recombined BACs were selected in kanamycin (25 μg/mL) and hygromycin (50 μg/mL).

### 3.2. ES Cell Targeting

First, 30 μg of the BAC-based targeting construct were digested with NotI, phenol-chloroform extracted, ethanol precipitated, resuspended in 30 μl of PBS and electroporated into HM-1 mouse embryonic stem cells [24] with a BioRad Gene Pulser (230 V, 500 μF). Two days after electroporation, ES cells were selected in neomycin (350 μg/mL) and hygromycin (160 μg/mL) until colonies appeared (7–10 days). ES cell colonies with normal morphologies were picked in 96-well plates, expanded and transgene copy number was estimated by Southern blot using a probe located 3′ downstream of the hygromycin cassette, which detects both the wt and transgene alleles 

### 3.3. FISH analysis of ES Clones

FISH was essentially performed as previously described [43]. In brief, BAC RPCI-23-362J7 DNA was labeled with digoxigenin-11-dUTP (Roche, Basel, Switzerland) by nick translation and for the detection of the probe sheep anti-digoxigenin-fluorescein isothiocyanate (FITC; Roche, Basel, Switzerland) and donkey anti-sheep-FITC (Dianova #713-095-147; Jackson Immuno Research, Cambridgeshire, UK) antibodies were used. Two hundred interphase nuclei per ES cell clone were scored for the number of FISH signals.

### 3.4. Quantitative Genomic PCR

Quantitative polymerase chain reaction (qPCR) was performed using the SYBR green method with 500 ng of genomic DNA isolated from ES cells, using the primers described in Figure 2C and Table 3. Genomic DNA levels were normalized for the murine *Rosa26* gene and relative genomic DNA abundance was calculated using the ΔCt (threshold concentration) method.

### 3.5. Animals

*Stat5/3^loxP/loxP^* mice (backcrossed five generation into C57BL/6N, purchased from Charles River, Wilmington, MA, USA), tetO-Cre mice (C57BL/6N) [30] and LAP-tTA mice (C57BL/6N) [31] were genotyped using primers P9/P11/P12, P15/P16 and P17/P18/P19, respectively. Experiments were performed using eight-week-old male mice. Mice were kept at the Decentralized Biomedical Facilities, Medical University of Vienna, under standardized conditions, and all animal experiments were carried out according to an ethical animal license protocol and contract approved by the Medical University of Vienna and Austrian Federal Ministry of Science, Research and Economy authorities (BMWF-66.009/0280-II/3b/2012). *Stat5/3^loxP/loxP^* strain will be available upon request.

*Stat5/3^Δ/+^* mice were generated by crossing *Stat5/3^loxP/loxP^* mice (backcrossed five generation into C57BL/6N) with mice expressing Cre recombinase under control of the Actb promoter [27], which were obtained from the Jackson laboratory (stock no. 003376) and back-crossed to C57BL/6N. Subsequently, the Actb-Cre transgene was removed by backcrossing with wt C57BL/6N mice. 

Blastocyst isolation was performed by superovulation and mating of heterozygous *Stat5/3^Δ/+^* females and males. Blastocysts were flushed from oviducts and uterus horns on E3.5 and cultured in M16 in an incubator for 24 h. DNA was extracted in 10 µl 1× homogenization buffer (containing 500 µg/mL Proteinase K), and genotyping was performed with the appropriated primers. 

### 3.6. Western Blot Analysis

Liver homogenates were prepared from snap frozen liver tissue. Protein lysates were blotted and probed with antibodies against HSC70 (sc-7298; Santa Cruz, CA, USA), STAT5A/B (rabbit polyclonal antibody, epitope aa775-788 and STAT3 (#06-596; BD Biosciences, Franklin Lakes, NJ, USA).

## 4. Conclusions

In conclusion, we have generated and validated a new mouse strain harboring a conditional allele for the *Stat5a*, *Stat5b* and *Stat3* loci. This new mouse strain will serve as a valuable tool to address the combined functions of STAT3 and STAT5A/B in tumorigenesis and other (patho)physiological processes, expanding the research tools of the JAK/STAT scientific community.

## Figures and Tables

**Figure 1 cancers-11-01226-f001:**
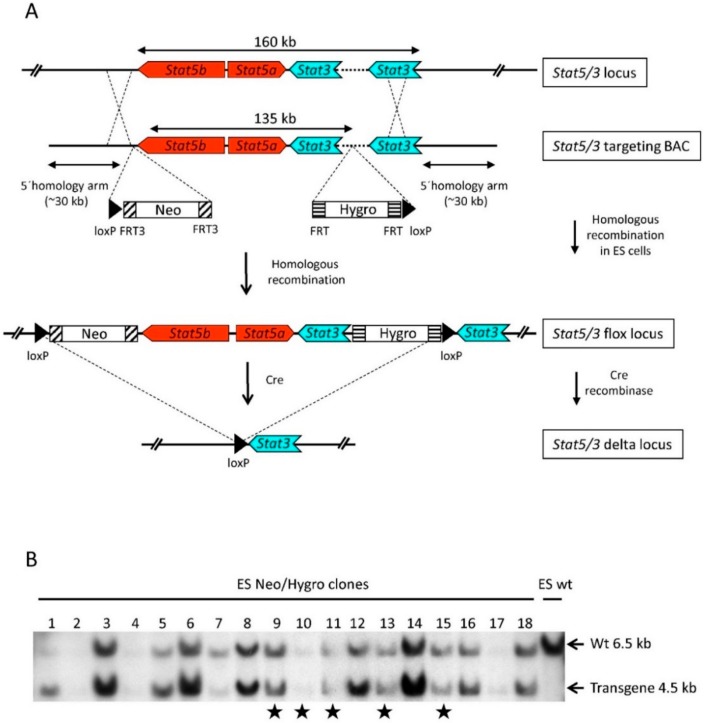
Conditional targeting the *Stat5a*, *Stat5b* and *Stat3* loci. (**A**) ES cells targeting strategy. A BAC spanning the *Stat5a/b* and *Stat3* loci was modified by introducing a neomycin resistance cassette flanked by FRT3 sites and containing a 5′loxP site upstream of the *Stat5b* locus. In a second recombineering step, a hygromycin resistance cassette flanked by two FRT sites and containing a 3′loxP site was introduced in the first intron of the *Stat3* locus (depicted as a dotted line), thus floxing the entire coding regions of the *Stat5a*, *Stat5b* and *Stat3* genes. (**B**) Southern blot analysis using *EcoRI*-digested genomic DNA from neomycin/hygromycin resistant ES cell clones and a probe localized 3′ downstream of the hygromycin cassette identified a band corresponding to the endogenous locus (wt, upper band) and the transgene (lower band). ES cell clones labeled with a star (9, 10, 11, 13 and 15) show roughly equal wt and transgene band intensities suggesting a correct targeting event or a random integration of two extra transgenic copies into the genome.

**Figure 2 cancers-11-01226-f002:**
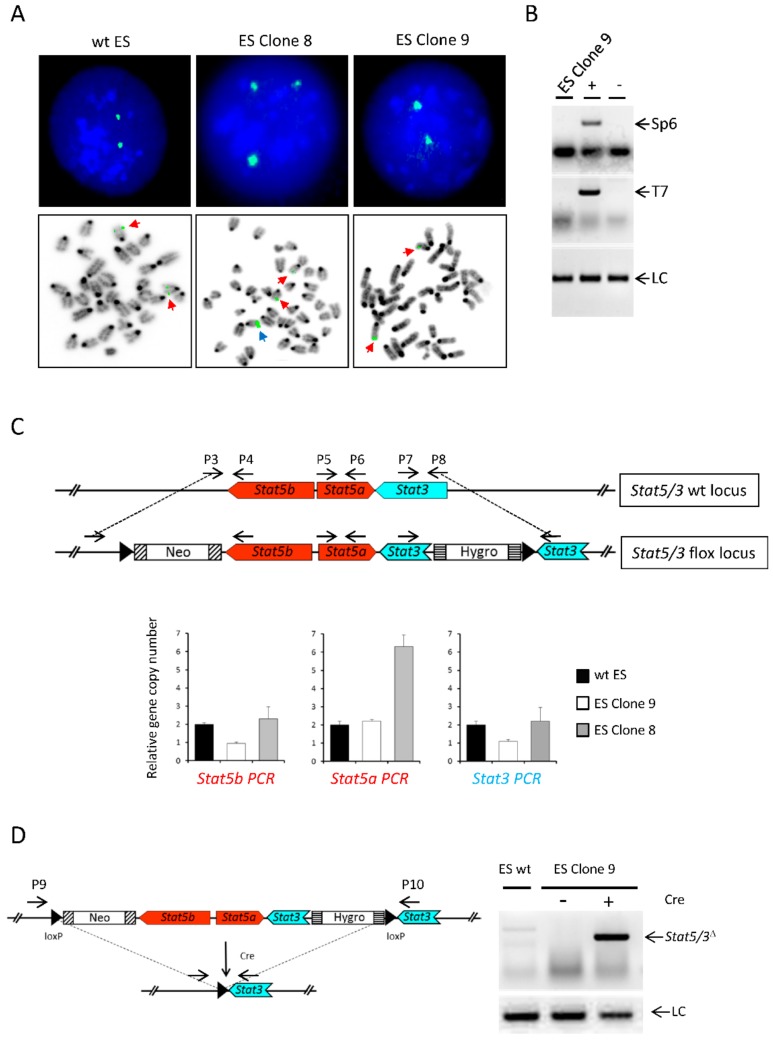
Identification of correctly targeted ES cells. (**A**) FISH analysis of wt ES cells and clones 8 and 9 using the RPCI-23-362J7 BAC as a probe showing that clone 9 has undergone correct homologous recombination. Upper pictures: representative interphase nuclei. Lower pictures: representative metaphases. Red arrows: endogenous locus, blue arrow: ectopic transgene integration. (**B**) PCR analysis using primers (P1/Sp6 and P2/T7), which amplify the ends of the BAC-based targeting construct, fails to detect the presence of BAC-ends in ES cell clone 9. +: BAC DNA (positive control). −: wt ES cell DNA (negative control) LC: loading control. (**C**) Quantitative PCR using primer pairs (P3/P4, P5/P6 and P7/8) spanning the *Stat5a/b* and *Stat3* loci shows that ES cell clone 9 harbors one copy of the *Stat5a/b* and *Stat3* endogenous locus and one copy of transgene in its genome (of note, primers P3/P4 and P7/8 do not amplify the transgenic construct). (**D**) PCR analysis (primers P9/P10) of genomic DNA from ES cell clone 9 electroporated with a plasmid expressing the Cre recombinase shows a specific product confirming the functionality of the loxP sites. LC: loading control.

**Figure 3 cancers-11-01226-f003:**
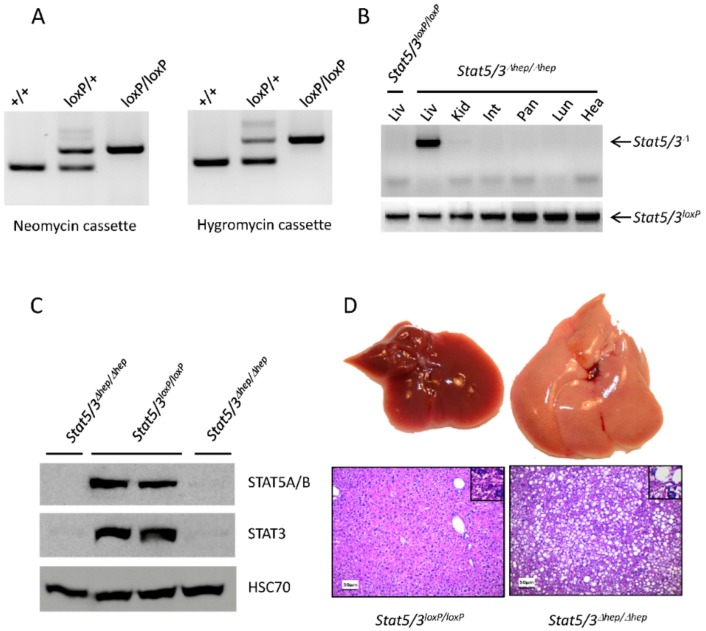
Validation of *Stat5/3^loxP/loxP^* mice. (**A**) PCR analysis of tail DNA using the primers P9/P11/P12 and P10/P13/P14 located in the region of the neomycin and hygromycin cassettes, respectively, showing the expected genotypes. (**B**) PCR analysis of genomic DNA extracted from different tissues revealed that the *Stat5a/b* and *Stat3* loci are deleted solely in the liver of *Stat5/3*^Δ*hep/*Δ*hep*^ mice (Liv: liver; Kid: kidney; Int: small intestine; Pan: pancreas; Lun: Lung; Hea: heart). (**C**) Western blot analysis using liver lysates from *Stat5/3^loxP/loxP^* and *Stat5/3*^Δ*hep/*Δ*hep*^ mice shows reduced levels of STAT5A/B and STAT3 protein in the livers of *Stat5/3*^Δ*hep/*Δ*hep*^ mice (HSC70: Heat Shock 70 kDa protein, loading control). Details can be found in Appendix A. (**D**) *Stat5/3*^Δ*hep/*Δ*hep*^ mice develop liver steatosis. Upper pictures: representative macroscopic liver images of control and *Stat5/3*^Δ*hep/*Δ*hep*^ mice. Lower pictures: representative liver sections from 8-week-old control and *Stat5/3*^Δ*hep/*Δ*hep*^ mice (*n* = 3) stained with hematoxylin and eosin (HE), scale bar: 50 µm.

**Table 1 cancers-11-01226-t001:** Quantification of the FISH analysis performed in the ES cell clones. FISH was performed using the RPCI-23-362J7 BAC as a probe in wildtype (wt) ES cells and clones 8, 9, 10, 11, 13 and 15. The result is expressed as the percentage of interphases (two hundred analyzed) containing one, two, three or four signals.

% of Cell Interphases				
ES Clone	One Signal	Two Signals	Three Signals	Four Signals
wt	1.5	73.0	20.5	5.0
8	0.0	9.0	85.0	6.0
9	4.5	93.5	2.0	0.0
10	0.0	3.0	27.5	69.5
11	0.0	4.0	83.5	12.5
13	0.0	11.0	80.0	9.0
15	0.0	1.0	29.0	70.0

**Table 2 cancers-11-01226-t002:** Genotype distribution of *Stat5/3* wt, heterozygote and homozygote animals. Number of embryos and pups from *Stat5/3^Δ^*^/+^ × *Stat5/3^Δ^*^/+^ breeding distributed accordingly to the genotype (expected number in brackets). Statistical differences from the expected numbers were analysed with the chi-square test using GraphPad QuickCalcs.

Genotype				
Stage	*Stat5/3* ^+/+^	*Stat5/3* *^Δ^* ^//+^	*Stat5/3* *^Δ^* ^/^ *^Δ^*	*p* Value
Blastocysts	8 (9)	23 (18)	5 (9)	0.1942
Born mice	22 (12)	27 (25)	0 (12)	<0.0001

**Table 3 cancers-11-01226-t003:** List of additional primers used in this study.

Primer	5′-3′Sequence
P1	CATCATACACTTCATTTTAGGACTGCC
P2	TGGCCCAGTGTTCAGTGCTCTTCTTACC
P3	GGGACTCTTAAAATGGAAATCTGG
P4	CAGAATGTTCTAGAAGGTTTGG
P5	GTGCATGCTTTGTAGGAATTCTATGG
P6	GTGGTTCCTCTGGTTTGTTACGTAGG
P7	ATCTTGGACACAAATGCAGAGCC
P8	CTGGTTAAGAAAAAGCCATTCTACC
P9	GCTTTGAAGCTTCATCCCTATCC
P10	TAGCTTAGGATAATTTTCTTCATG
P11	GGTTGGCGCCTACCGGTGGATGTGG
P12	CAGTAGCCCAGTGTCCCAGCCAAACAG
P13	AATCAGTAATAAGTGATGATAGAAGGG
P14	CAGATGACCACTCCAGTCGGGGG
P15	ACCAGCCAGCTATCAACTCG
P16	TTACATTGGTCCAGCCACC
P17	TCTGAGCATGGCCTCTAA
P18	GCTGGAGTAAATTTCACAGTG
P19	TCTCACTCGGAAGGACAT
Sp6	GATTTAGGTGACACTATAG
T7	GTAATACGACTCACTATAGGGC

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
