# Peer review of "A Mouse Model to Assess STAT3 and STAT5A/B Combined Inhibition in Health and Disease Conditions"

_cancers, 2019, doi:10.3390/cancers11091226_

Round 1

Reviewer 1 Report

The authors revised the manuscript according to my comments.

The editor(s) should decide if the description of a new mouse line is sufficient to justify publication in MDPI Cancers.

Reviewer 2 Report

All suggested concerns have been addressed.

This manuscript is a resubmission of an earlier submission. The following is a list of the peer review reports and author responses from that submission.

Round 1

Reviewer 1 Report

The manuscript by Moll et al. describes the generation of mice harboring a flox Stat3-Stat5a/b allele. The corresponding genes are localized within a 160 kb region on chromosome 11. Therefore, it would be virtually impossible to generate a combined deletion of Stat3 and Stat5a/b by intercrossing Stat3 and Stat5a/b single flox mice. Homozygous mice lacking the Stat3-Stat5a/b region in all tissues are not viable, similar to Stat3 knock-out mice. Mice with a hepatic deletion of the chromosomal region develop steatosis, similar to mice lacking both STAT5 isoforms in the liver. The authors discuss that the newly generated strain will be a valuable tool to evaluate therapeutic strategies targeting both STAT3 and STAT5 isoforms.

The manuscript is well written, the experiments seem to be performed very well, many details on the generation of the novel mouse strain are given.  This study is of interest for scientists working on the Jak/Stat pathway and on GEMMs, in particular those planning the generation of mice (conditionally) lacking STAT3, STAT5A and STAT5B in certain tissues.

The following points should be addressed:

A more thorough characterization of the Stat3/5Dhep mice would strengthen the manuscript:

(A)   The authors should investigate the activation of other STAT factors (e.g. upon IL-6 or GH stimulation) which may be hyperactivated in the absence of STAT3, STAT5A and STAT5B. Such compensatory activation of “other” STAT members in absence of a certain STAT factor has often been observed which renders the interpretation of the observed phenotypes difficult. This disadvantage of knock-out approaches should also be discussed, next to the advantages which are being mentioned in the introduction and discussion section.

(B)   Mice lacking STAT3 selectively in the liver show distinct phenotypes, e.g. increased susceptibility to fibrosis, decreased mitosis upon partial hepatectomy, or effects on metabolism. How do the newly generated Stat3/5Dhep mice compare to mice lacking only STAT3 in hepatocytes?

Minor points:

-        Targeting strategy schemes (Figs. 1, 2): The position of the probe and of the EcoR1 sites should be shown in the scheme. The sizes of the EcoR1 fragments (Fig. 1B) should be indicated.

Legend Fig. 1: a probe localized upstream of the hygromycin cassette, l. 231: using a probe located upstream of the stat5b gene – does this refer to the same probe?
Does “upstream” mean relative to the transcription of the down-stream gene?

-        Is a truncated version of STAT3 made? If yes, how long would that be?

-        On what day do the Stat5/3DD embryos die?

-        L. 193: Do the authors have evidence that other Cre transgenic lines may not be as efficient to delete such a large genomic region (135 kb) flanked by loxP sites?

-        Fig. 2A, FISH analysis, blue/red arrows: how is the differentiation made between ectopic transgene expression and integration into the endogenous locus?

-        l. 144: …born at Mendelian ratio (figure 3A) – not shown in Fig. 3A…

-        There are several minor mistakes: l. 59: resultS, l. 73: demonstrateS, l. 112 manly, l. 140: probe, l. 175: matings, l. 261: genotyping was performed, table 3: Stat5/3++

Reviewer 2 Report

Please correct line 144- "Stat5/3loxP/loxP mice are viable, born at Mendelian ratio (figure 3A)", should refer to Table 2.

Could the authors add some phenotypic studies to elucidate further the functional sinificance of deleting STAT5/STAT3 in liver? It is known that STATs affect recovery after liver injury, cause liver fibrosis and affect proliferation. These studies would further exemplify the utility of this unique model in future studies.

Reviewer 3 Report

The manuscript by Herwig Moll et al. deals with the generation of a novel transgenic mouse model for combined conditional STAT3/STAT5A/B-deletion. Taking advantage of the close proximity of the 3 genes STAT3, STAT5A, and STAT5B on mouse chromosome 11 (https://genome.ucsc.edu/cgi-bin/hgTracks?db=mm10&lastVirtModeType=default&lastVirtModeExtraState=&virtModeType=default&virtMode=0&nonVirtPosition=&position=chr11%3A100657344%2D101131922&hgsid=738448267_xHH85Jee1zyBSrKiKVOQrtqEc5lU), the authors flanked the gene region with floxP sites. Using a mouse line that expresses cre recombinase in a variety of cell types/tissues (based on the Actb promoter), the authors show that the combined deletion of STAT3 and STAT5A/B is embryonic lethal - as was expected from the described phenotypes of the single deletions. A liver-specific Cre expression (using a cross of the tetO-Cre mice and LAP-tTA line) resulted in efficient ablation of the STAT3 and STAT5A/B proteins in the liver. As expected from the previously known phenotype of the STAT3-deficient mouse, the combined loss of STAT3 and STAT5A/B results in pronounce lipid accumulation in the liver.

The generation of the new mouse model is well executed and controlled.

The paper is well written and easy to follow. The experiments are convincing. The claims are justified by experimental data.

Major point

Unfortunately, I miss scientific insights. What do we learn from the study? The reader wonders why they undertook such efforts without some keen-witted functional tests in mind. The authors state that the mouse may be a future model for anti-tumor drugs targeting STAT3 and STAT5A/B, but I never heard that lead compounds are often compared to highly complex conditional knockout mouse models.

In my opinion, the authors should apply their new model to generate relevant functional data.

Minor point

There is an ongoing debate concerning 129 strain derived passenger mutations confounding interpretation of genetically modified mice (see e.g. https://www.ncbi.nlm.nih.gov/pmc/articles/PMC4800811/). This issue should be discussed.

The authors state “Stat5a/b and Stat3 are genetically linked and located within an approximately 160 kb region of the mouse genome. Consequently, combined deletion of Stat5a/b and Stat3 by breeding currently available mice harboring conditional alleles for Stat5a/b and Stat3 is virtually impossible” (see end of page 6). Yet, others claim that they actually did it (see https://academic.oup.com/endo/article/154/7/2434/2423363)

The auhtors state "However, other Cre transgenic lines may not be that efficient" concerning the deltion. It sounds like the authors tried other deleter lines. What was their experience?
